# Adenoviral Pneumonia Outbreak in Immunocompetent Adults—A Missed Antimicrobial Stewardship Opportunity?

**DOI:** 10.3390/antibiotics14010023

**Published:** 2025-01-02

**Authors:** Branimir Gjurasin, Lorna Stemberger Maric, Tvrtko Jukic, Leona Radmanic Matotek, Snjezana Zidovec Lepej, Marko Kutlesa, Neven Papic

**Affiliations:** 1Department for Intensive Care, University Hospital for Infectious Diseases “Dr. Fran Mihaljević”, 10000 Zagreb, Croatia; bgjurasin@bfm.hr (B.G.); tjukic@bfm.hr (T.J.); mkutlesa@bfm.hr (M.K.); 2Department for Infectious Diseases, School of Medicine, University of Zagreb, 10000 Zagreb, Croatia; lmaric@bfm.hr; 3Department for Pediatric Infectious Diseases, University Hospital for Infectious Diseases “Dr. Fran Mihaljević”, 10000 Zagreb, Croatia; 4Department for Immunological and Molecular Diagnostics, University Hospital for Infectious Diseases “Dr. Fran Mihaljević”, 10000 Zagreb, Croatia; lradmanic@bfm.hr (L.R.M.); szidovec@bfm.hr (S.Z.L.); 5Department for Viral Hepatitis, University Hospital for Infectious Diseases “Dr. Fran Mihaljević”, 10000 Zagreb, Croatia

**Keywords:** adenovirus, pneumonia, viral pneumonia, ARDS, cytokine

## Abstract

**Background/Objectives**: While the concept of viral community-acquired pneumonia (CAP) changed with COVID-19, the role of non-influenza viruses as a cause of CAP is less clear. The aim of this study was to describe the clinical course, risk factors, inflammatory profiles, antibiotic use, outcomes and complications of adenoviral (AdV) CAP. **Methods**: A prospective, non-interventional, observational cohort study included consecutively hospitalized immunocompetent adult patients with AdV CAP during an 18-month period. Clinical and laboratory data, including lymphocyte subpopulations and serum cytokine profiles were collected and correlated to clinical outcomes. **Results**: Fifty-eight patients with AdV CAP were included; 81% were males, with a median age of 33 (IQR 28–41) years and 62% without any comorbidities. All patients initially had high-grade fever for a median duration of 6 (5–7) days and respiratory symptoms. Increased CRP and procalcitonin, lymphopenia, mild thrombocytopenia and liver injury were frequent. Radiographic findings mimicked bacterial pneumonia (83% had unilateral involvement). Twenty-two patients (38%) had criteria for severe CAP, and these patients had higher procalcitonin, NLR, AST, ALT, LDH and CK, and lower T-lymphocyte CD4+ count. In comparison to influenza and bacterial CAP, patients with AdV had higher serum IL-2, IL-1β, IL-8, IL-10, CXCL10 and MCP-1, and lower TGF-β1 concentration. Thirteen patients required low-flow oxygen therapy, and 13 advanced respiratory support. Complications occurred in 29%, with one fatal outcome. While all patients received empirical antibiotic therapy, after AdV detection it was stopped in 21%, although only one patient had detected a possible bacterial coinfection. **Conclusions**: Since AdV CAP in immunocompetent patients is clinically and radiologically indistinguishable from bacterial CAP, it is associated with prolonged clinical course and lack of clinical response to antibiotics. This emphasizes the importance of AdV testing which could lead to more rational antimicrobial treatment.

## 1. Introduction

Community-acquired pneumonia (CAP) is one of the most common infectious diseases globally, one of the most common reasons for prescribing antibiotics, one of the leading causes of hospitalization and death, and the most common cause of acute respiratory distress syndrome (ARDS) and sepsis, with high mortality that has not declined over time [1,2,3]. Most patients with CAP do not have an identified etiology. In a pivotal EPIC study, despite a comprehensive microbiologic work-up, the pathogen was detected in only 38% of patients. Notably, 23% were viral, and 11% were bacterial, raising the question could viral CAP be more common than bacterial CAP in adults [4]. In a related study, viruses were more frequently detected in upper respiratory samples in patients with CAP than in asymptomatic controls, with rare viral detection in asymptomatic adults when compared to children [5]. The COVID-19 pandemic further demonstrated the importance of non-influenza viral pneumonia in immunocompetent adult patients. With the wider availability of multiplex PCR tests for respiratory pathogens with increased diagnostic yield compared to viral culture and antigen detection assays, the concept of viral CAP in the adult population significantly changed [6,7,8].

Adenovirus (AdV) has been reported to cause pneumonia, mostly in children and immunocompromised adults, with no seasonal distribution. In adult immunocompetent patients, AdV is a rare cause of CAP, with the majority of data on clinical course from outbreaks in military recruits. AdV CAP is often characterized by focal and unilateral lung infiltrates, elevated inflammatory parameters, and persistent fever that may mimic typical bacterial CAP, a finding not frequently observed in other viral infections [9,10,11]. Furthermore, taking the well-known association of influenza CAP with bacterial coinfection into account, AdV CAP is frequently treated with antibiotics, although there are no well-documented reports of increased frequency of bacterial coinfection. The resulting uncertainty also arises from the lack of data describing the clinical course, complications, and outcomes of AdV CAP. Since AdV CAP has rare occurrences and is rarely suspected outside outbreaks, it is frequently misdiagnosed and managed as bacterial CAP.

Here, we describe an outbreak of AdV CAP in immunocompetent adults from the general, non-military population in Croatia who required inpatient management during an 18-month period. We describe demographic, clinical, and laboratory data, including lymphocyte subpopulation and serum cytokine analysis, outcomes, and complications during hospitalization. Since clinical and laboratory features of AdV are not distinguishable from bacterial CAP, an expanded inflammatory cytokine panel was compared to other CAP pneumonia etiologies and correlated with clinical outcomes. We also compare the mentioned data between patients with non-severe and severe CAP (sCAP). Finally, we discuss the importance of AdV detection on antimicrobial stewardship of CAP, specifically the more judicious use of antibiotics in this patient group.

## 2. Results

### 2.1. Clinical Characteristics of Hospitalized Patients with AdV Pneumonia

Overall, 58 patients with AdV pneumonia were hospitalized during the study period and included in this study. The majority of patients were males (47, 81%) with a median age of 33 (28–41) years and without any comorbidities (37, 61.7%). Twenty patients (34.5%) were smokers, and 13 (22.4%) reported moderate alcohol consumption. No patient reported recently experiencing crowded environments, including mental health facilities, job training sites, boarding schools, prisons, or military sites.

All patients initially showed influenza-like symptoms and had high-grade fever with a peak body temperature of 39.8 (39.2–40.0) °C for a median duration of 6 (5–7) days before hospital admission. Respiratory symptoms were present in the majority of patients, predominantly dry cough (60%), and less frequently, catarrhal symptoms (33%) or sore throat (41%). A significant number of patients had diarrhea (69%), as presented in Table 1.

Before hospital admission, 34 (58.6%) of patients received antibiotics for a median duration of 3 (1–5) days. Eight patients (13.8%) were treated with beta-lactam, eleven (19%) with macrolide, eleven (19%) with macrolide + beta-lactam combination therapy, and one with a respiratory quinolone.

Regarding disease severity at admission, 22 (37.9%) patients were classified as sCAP according to IDSA criteria, and differences in baseline patients’ characteristics between these groups are shown in Table 1.

As expected, compared to patients with non-severe CAP, patients with sCAP more frequently reported tachypnea and dyspnea and had higher respiratory frequency and lower spO2/FiO2 ratio. Median time to hospital admission was longer in sCAP (8, IQR 7–10 vs. 6, IQR 4–6 days). There were no differences in age, sex, comorbidities, and other clinical symptoms and signs between the groups (Table 1).

Regarding laboratory findings on admission, the majority of patients had elevated inflammatory markers C-reactive protein (CRP, 141, IQR 91–236 mg/L), procalcitonin (0.46, IQR 0.22–1.59 µg/L) and fibrinogen (5.6, IQR 5.2–6.3 g/L), normal white blood count (WBC, 6.3, IQR 5.0–8.5 × 10^9^/L) but with frequent lymphopenia (relative number of lymphocytes 13%, IQR 9–18% and absolute number of 771 cells/μL, IQR 580–1072/μL) and mild thrombocytopenia (138, IQR 116–165 × 10^9^/L). The neutrophils–lymphocytes (NLR) ratio and lymphocyte–monocyte ratio (LMR) were lower than the reference range, while the platelet–lymphocyte ratio (PLR) was higher (Table 2). Mild hepatocellular injury was common (AST 63, IQR 32–105 IU/L and ALT 35, IQR 27–59 IU/L), as well as an increase in lactate dehydrogenase (LDH, 364, IQR 246–665 IU/L) and creatine kinase (CK 604, IQR 174–1052 IU/L).

As shown in Table 2, patients with sCAP had significantly higher procalcitonin levels (but not CRP) and lower NLR. Liver injury (as measured by AST, ALT, GGT), LDH, and CK, as well as kidney injury (AKI) was significantly greater in patients with sCAP.

Of the 22 patients with sCAP, 16 had AdV detected in the blood (viremia), and 7 patients with successful serotyping of AdV in respiratory samples had confirmed AdV-7 serotype infection.

Twenty-two patients had a flow cytometry analysis of lymphocyte subpopulations in peripheral blood, as presented in Table 3. Relative T-cell lymphopenia (CD3+ < 60%) was observed in 5 (22.7%), B-lymphocytosis (CD19+ > 20%) in 13 (59.1%) and decreased absolute CD4+ count in 16 (72.7%) with a median of 303 (178–569) cells/μL, and decreased CD4+ percentage (<35%) in 8 (36.4%) patients. Regarding cell activation markers, CD8+CD38+ cells (>7%) were increased in 16 (72.7%) and HLA-DR+ (>10%) in 12 (54.5%) patients. As shown in Table 3, patients with sCAP had significantly lower absolute CD4+ counts, but there were no differences in other lymphocyte populations, which were already disturbed, as previously mentioned.

All patients had chest X-rays performed on admission. The majority of patients had unilateral involvement (48, 82.76%), with only one lobe affected in 30 (51.7%). Two lobes were affected in 13 (22.4%), 3 in 5 (8.6%), 4 in 3 (5.2%) and all 5 in 7 (12.1%). Ten patients (17.2%) had bilateral involvement and 32 (55.2%) had pleural effusion. As expected, patients with sCAP more frequently had bilateral pneumonia at hospital admission (9, 40.9% vs. 1, 4.5%) and ≥ 3 lobes affected (15, 68.1% vs. 0). In Figure 1, we present examples of characteristic radiologic findings in our patients which are radiographically indistinguishable from bacterial pneumonia.

### 2.2. Clinical Course and Outcomes of AdV Pneumonia

During hospitalization, 32 (55.2%) patients did not require oxygen therapy, 13 (22.4%) required low-flow (≤25 L/min) oxygen therapy for a median duration of 4 (IQR 3–8) days. Thirteen patients (22.4%) required advanced respiratory support; of them, two were treated with high-flow nasal oxygen therapy (HFNOT, 3.4%), one with non-invasive mechanical ventilation (NIV) and ten with invasive mechanical ventilation (IMV), of whom six were treated with venovenous extracorporeal membrane oxygenation (vv-ECMO) for a median duration of 6 (IQR 5–7) days, as shown in Figure 2. Patients who were treated with IMV required prolonged ventilation of 21 (IQR 14–33) days.

On day 5 of hospitalization, 16 (27.59%) patients were discharged, and 42 patients remained hospitalized, of whom 20 (34.48%) required no oxygen supplementation, 13 (22.4%) standard oxygen therapy (up to <25 L/min), 1 HFNOT, 5 (8.6%) IMV and 3 (5.17%) were on vv-ECMO. We also examined the kinetics of routine laboratory markers on day 5 of hospitalization, and the majority of our patients still had elevated CRP (109, IQR 54–138), procalcitonin (0.32 µg/L, IQR 0.19–0.89), AST (92 IU/L, IQR 38–132), ALT (80 IU/L, IQR 38–114) and LDH (400 IU/L, IQR 260–787), as well as lymphopenia (21%, IQR 14–25%) and decreased NLR (3.1, IQR 2.3–5.6). In RM-ANOVA analysis, to compare laboratory parameters kinetics between patients with sCAP and non-severe CAP, only PCT and WBC significantly differed on day 5 between the two groups, as shown in Figure 3.

As already presented, patients with AdV had prolonged clinical course, with a median duration of fever of 9 days (IQR 8–10), and a median duration of hospitalization of 6 days (IQR 4–10). Patients with sCAP had a significantly longer duration of hospitalization, with a median of 14 days (IQR 9–34).

At admission, only one patient had a microbiologically confirmed possible bacterial coinfection (*S. pyogenes* in throat swab). Regarding other respiratory viruses, only one patient had microbiologically confirmed coinfection (RSV). All patients after endotracheal intubation had bacterial and fungal cultures performed, and in two patients *Aspergillus fumigatus* was isolated, and invasive aspergillosis was later confirmed with positive galactomannan test in bronchoalveolar fluid (BALF) and blood.

All patients received empirical antibiotic therapy for a median duration of 7 (5–10) days, mostly beta-lactam and macrolide combination (28, 48.2%) and beta-lactam monotherapy (13, 22.4%). None of the patients received cidofovir or brincidofovir, and 16 (27.59%) patients received short-course corticosteroids. After AdV detection, antimicrobial therapy was stopped in 12 (20.69%) patients; 9 patients with non-severe CAP and 3 with sCAP.

Overall, 17 (29.3%) patients had complications during hospitalization. Eight patients (13.8%) developed AKI, and of them six required renal replacement therapy (median duration of 8, IQR 5–26) days. Two patients had pneumothorax. Nine (15.2%) developed nosocomial infections, most commonly urinary tract infections (three patients) and ventilator-associated pneumonia (four patients). In our cohort, none of the patients had confirmed acute heart failure or pulmonary thrombosis. One patient died during hospitalization.

### 2.3. Analysis of Cytokine Profiles in sCAP Caused by AdV, as Compared to Influenza and Bacterial Pneumonia

Next, serum concentrations of 13 cytokines were measured in patients with AdV sCAP and compared to patients with sCAP caused by influenza H1N1 (from season 2023/2024) and bacterial pneumonia. As presented in Figure 4, patients with AdV sCAP had high expression of all measured proinflammatory cytokines. Different cytokine profiles were observed compared to influenza and bacterial pneumonia, with IL-2, IL-1β, IL-8, IL-10, CXCL10, and MCP-1 being highest in the AdV group, while free TGF-β1 was lowest in the AdV group. There were no differences in serum concentrations of IL-4, IL-6, IL-17A, TNFα and IL-12p70.

Next, we analyzed cytokine concentrations in patients with AdV pneumonia who had criteria for acute respiratory distress syndrome (ARDS) and found significantly higher serum concentrations of CXCL10 (9629 pg/mL, IQR 6130–12,779 vs. 4265 pg/mL, IQR 1749–5499) and MCP-1 (832 pg/mL, IQR 695–2133 vs. 226, IQR 506–198) and lower of TGF-β1 pg/mL (79 pg/mL, IQR 70–106 vs. 150, IQR 131–157), as shown in Figure 5.

## 3. Discussion

Here, we describe clinical and laboratory characteristics of hospitalized adult immunocompetent patients with AdV CAP during a 2023–2024 community-based outbreak in Croatia. Patients were predominantly previously healthy males, without obvious risk factors, even in patients with sCAP, presenting with prolonged high-grade fever, dry cough, and diarrhea. We identified no epidemiological links among patients. Most patients received antibiotics prior to hospital admission (i.e., prior to AdV detection). Notably, clinical, laboratory, and radiological findings mimicked bacterial CAP. Patients had a prolonged clinical course; 38% had sCAP, 22% required standard oxygen therapy, and 23% advanced respiratory support. Despite prolonged IMV and duration of hospitalization, patients had good outcomes with full recovery.

Serious AdV infections have mostly been reported in immunocompromised patients in whom they cause a wide spectrum of disease, from asymptomatic viremia to disseminated infection associated with organ failure and death [12]. In children, the concept of viral etiology of CAP, including AdV, has been established for decades [13,14]. In adults, AdV CAP has mainly been reported during epidemics in military recruits [15,16,17], which are difficult to compare to the general adult population since they live in extremely close quarters, are without comorbidities, are physically active, and have more accessible healthcare, and consequentially shorter time from symptoms onset to medical evaluation. Published case series of military recruits with AdV CAP (serotypes 4, 7, 14, and 55), mostly from the US and Southeast Asia, report a high prevalence of cough, prolonged fever, normal WBC counts or leukopenia with monocytosis and lymphopenia, thrombocytopenia, elevated CRP, mild hepatocellular injury, and radiographical lobar pneumonia [16,17,18,19,20,21,22], similarly as in our cohort. Fatalities were rarely reported; however, their true rate is difficult to establish due to possible publishing bias.

In our cohort, lymphopenia and mild thrombocytopenia were frequent findings in AdV patients, similarly to other viral infections [23]. In addition, hematological ratios (NLR, MLR, and PLR) were altered. Adequate T-cell responses are crucial to control AdV infection, as shown in immunocompromised patients that have the highest burden of the disease with the highest mortality rates [24]. Furthermore, we observed significantly altered lymphocyte subpopulations, implying possible immune system imbalance even in immunocompetent adults. These included decreases in CD3+, CD4+, and NK cells and a relative increase in B lymphocytes. Absolute CD4+ cell count correlated with disease severity, and the majority of T-lymphocytes expressed markers of activation. Similarly, in the pediatric population, it was suggested that the malfunction of T-cells might be due to the inflammatory response and that T-cell depletion might be associated with mortality [25,26]. Our results suggest that monitoring lymphocyte subpopulations, specifically CD4+ count, has clinical significance in assessing the severity of pneumonia.

Next, patients with AdV CAP had substantially elevated inflammatory markers (CRP, PCT, fibrinogen), laboratory features that are not distinguishable from bacterial CAP, as previously reported [9,11]. Therefore, an expanded inflammatory cytokine panel was compared to influenza and bacterial CAP. All measured cytokines were markedly elevated in all three groups, but they had distinct expression patterns. While the concept of “cytokine storm” regained interest with COVID-19, it is not specific to SARS-CoV2 and has been linked to other infectious syndromes, including viral pathogens [27,28]. However, the question of how cytokine networks are shaped and regulated in different pneumonia contexts remains unanswered [29]. Only a few studies examined the association of inflammatory cytokine levels with different AdV disease severity. The largest study was from two military outbreaks in China and included 65 patients with upper respiratory tract infection and 95 patients with AdV CAP and reported a positive correlation of serum levels of CXCL10, IL-2, and TNF-α with disease severity, and reduced levels of IFN-γ and IL-10 in non-severe CAP patients [30]. A retrospective cohort study from China explored differences in cytokine concentrations among patients with ARDS caused by AdV, influenza A/pH1N1, and bacterial pneumonia, and found higher expression of MCP-1, TNF-α, SDF-1α, IL-17, SCF-β, and TRAIL in the AdV group [31]. Yet, cytokine expression is regulated by a variety of factors, including age, comorbidities, genetics, race, individual cytokine–receptor interactions, or environmental factors, which explains the observed differences between the studies and might not necessarily be generalized to other populations. In our cohort of Caucasian patients outside military settings, patients with AdV sCAP had higher serum IL-2, IL-1β, IL-8, IL-10, CXCL10 and MCP-1, and lower TGF-β1 concentrations. Furthermore, CXCL10, MCP-1, and TGF-β1 correlated with ARDS.

IL-2, a regulatory Th1 cytokine, is mainly secreted by activated CD4+ and CD8+ cells and has recently been linked with the severity and complications of CAP by enhancing inflammation and promoting cytokine production [32]. IL-8 is a potent neutrophil chemotactic factor shown to be associated with the development of ARDS, severity of sepsis, and a prognostic factor of COVID-19 [33,34]. The anti-inflammatory IL-10 is an important regulator of the immune response to bacterial pneumonia which suppresses the ability of neutrophils to kill *S. pneumoniae* and regulates lung inflammation [35]. In AdV, similarly as reported in COVID-19, an increase in IL-10 might indicate a failed attempt to suppress the hyperinflammatory response [33]. CXCL10, a proinflammatory chemokine involved in the recruitment of leukocytes, can create local amplification loops responsible for sustaining lung inflammation [36]. Interestingly, serum concentrations of anti-inflammatory TGF-β1 were lower in AdV than in influenza and bacterial pneumonia and even lower in a subgroup of patients with ARDS. In the context of influenza A infection, TGF-β1 has a strong pro-viral effect by suppressing early immune response during infection [37]. In COVID-19, TGF-β1 was significantly increased and correlated with disease severity and outcomes [38]. In ARDS, uncontrolled activation of TGF-β1 was implicated in the development of lung fibrosis [39].

Hence, these cytokines might have important roles in the pathophysiology of AdV pneumonia and its complications that warrant further exploration. Currently, there is no recommended treatment for AdV pneumonia in immunocompetent adults. While corticosteroids and immunomodulators might be attractive options, as in COVID-19, where they became standard of care, their role and impact on AdV outcomes have yet to be defined. A better understanding of AdV inflammatory responses might lead to the development of new therapeutic strategies.

Compared to the recently published study including 102 AdV (50% AdV-55 and 29% AdV-7) CAP immunocompetent adult patients from China [9], our cohort had lower SOFA scores, less frequently required oxygen and advanced respiratory support, less frequently progressed to ARDS (21% vs. 40%), and had shorter hospital stays. This could be explained by the hospitalization of patients with less severe CAP in our cohort and possibly due to higher virulence of AdV-55, which was not detected in our outbreak [9]. Although we detected only serotype 7, our results might correspond to other serotypes due to the similarities with other published cohorts. A study describing an older cohort (46 years, IQR 31–56) with more comorbidities reported more frequent ICU admittance, a longer hospital stay, and a higher mortality (4% vs. 1.7%), suggesting the impact of age and comorbidities on patient outcomes [11].

In our study, all patients received empirical antibiotic therapy, which was stopped in only 21% of patients after AdV detection despite no evidence of bacterial co-infections. The low rate of stopping antibiotics suggests the uncertainty that CAP is solely caused by AdV, in part due to bacterial-CAP-mimicking features. Furthermore, current CAP management guidelines recommend antibiotic treatment for every non-COVID-19 and non-influenza CAP; however, they do not specifically address AdV [40,41]. We are not aware of studies examining antibiotic de-escalation or withholding in AdV CAP patients. However, as we are becoming more familiar with the high prevalence and clinical features of viral CAP in adults, the benefits of antibiotic de-escalation strategies are more recognized, specifically since bacterial co-infections are rare in these settings [42,43]. Furthermore, the importance of viral PCR testing for non-influenza, non-SARS-CoV-2 viruses in sCAP patients is becoming apparent, leading to recommendations on their use in routine clinical practice [6,42,43,44].

A combination of clinical and laboratory features of CAP with results of microbiology work-up, including PCR testing for respiratory viruses, could contribute to more rational antibiotic use—not escalating antibiotics in cases of persistent fever and deescalating or even stopping antibiotics in case of viral CAP. This could be an important opportunity for lowering treatment costs, length of stay, risk of adverse events, and potential antimicrobial resistance. Several studies found similar clinical outcomes between patients with viral pneumonia who continued antibiotics after virus recognition and those who did not, suggesting the opportunity for de-escalation of empiric antibiotic therapy when a virus is identified [45,46,47]. Although our study was not designed to examine the impact of antibiotic therapy on AdV CAP outcomes, our findings support withholding or stopping antibiotic therapy in patients with AdV-positive CAP without evidence of bacterial infection if patients are clinically stable. In the case of AdV CAP, radiologic characteristics of lung infiltrates and elevated inflammatory markers do not seem to have a discriminatory role in bacterial infection.

Our study should be viewed within its limitations; the patients were from one AdV outbreak, so a situation bias is possible; a relatively small proportion of patients were tested for viremia with a qualitative test at one point, so dynamic changes in viremia could not be analyzed; only serotype AdV-7 was detected, so our results might not correspond to infections with non-AdV-7 serotype. Due to the study design, lack of control groups, and the fact that most patients received antimicrobial therapy, the analysis of the association of antimicrobial treatments with patients’ outcomes could not be performed and this causation could not be established. There is also a risk for referral bias regarding sCAP patients, as our hospital serves as a national referral center for extracorporeal oxygenation.

Nevertheless, we studied a well-defined cohort of patients hospitalized with AdV CAP, described their clinical course and outcomes, and provided important clinical and laboratory clues that might have practical implications.

## 4. Materials and Methods

### 4.1. Study Design, Patients and Samples

This was a prospective, non-interventional, observational cohort study that included all immunocompetent adult patients consecutively hospitalized with AdV CAP between January 2023 and July 2024 at the University Hospital for Infectious Diseases (UHID) Zagreb, Croatia. Inclusion criteria were diagnosis of CAP suggested by at least two of the following: cough, purulent sputum, chest pain, dyspnea, fever; the presence of lung shadowing/infiltrate on chest X-ray or CT-scan at admission or during the 48 h post-hospital admission and a positive PCR test for AdV on a respiratory sample (nasopharyngeal and/or oropharyngeal swab, endotracheal aspirate or BALF). Exclusion criteria were any of the following: age < 18 years, immunocompromise, active neoplastic disease, active autoimmune diseases, clinical history suggesting aspiration of gastric content, SARS-CoV-2 infection in the last 90 days, or tuberculosis.

All patients were tested with multiplex PCR for respiratory pathogens routinely with viral multiplex panel (which included AdV, human metapneumovirus—hMPV, RSV, human rhinovirus—hRV, SARS-CoV-2, influenza and parainfluenza virus—PIV), while a subset of patients with sCAP in the ICU was tested with bacterial/viral multiplex panel on upper respiratory samples (bioMérieu BioFire FilmArray Respiratory Panel, Marcy-l’Étoile, France, which detects AdV, influenza virus, RSV, hMPV, PIV, seasonal coronaviruses, hRV or enterovirus, SARS-CoV-2, *Chlamydia pneumoniae* and *Mycoplasma pneumoniae* and *Bordetella pertussis*) and/or on BALF sample (bioMérieu BioFire FilmArray Pneumonia plus panel, Marcy-l’Étoile, France, which detects 27 bacterial and viral respiratory pathogens, including AdV, influenza virus, RSV, hMPV, PIV, seasonal coronaviruses, hRV or enterovirus, MERS-CoV, *Chlamydia pneumoniae, Mycoplasma pneumoniae* and *Legionella pneumophila*). Patients with sCAP had their blood tested for AdV viremia, as well as a second respiratory sample on which serotyping was attempted.

Patients who entered the study had their serum sampled on the first day of hospitalization for cytokine concentration analysis. The samples were divided into aliquots to avoid repeated freeze/thaw cycles and stored at −80 °C until testing.

Serum samples from 12 patients with sCAP caused by infection with influenza A/pH1N1 from season 2023/2024 and 12 patients with bacterial pneumonia collected in 2024 were included for the analysis of cytokine profiles.

The study conformed to the ethical guidelines of the Declaration of Helsinki and was approved by the School of Medicine, University of Zagreb Ethics Committee (protocol code 01-1247-2-2019, date of approval 30 August 2019). All participants gave written informed consent.

### 4.2. Data Collection and Definitions

Routine clinical and demographic data were collected: gender, age, comorbidities, chronic medication use, days from symptom onset on admission, the severity of pneumonia (SOFA, PSI, CURB-65, SMART-COP scores), routine laboratory blood studies on admission and on the fifth day of hospitalization (as presented in Table 2), and routine microbiologic work-up, including blood cultures, Legionella urinary antigen, urine cultures, endotracheal, BALF and pleural fluid culture (including serum and BALF fungal studies in IMV patients with prolonged or progressive clinical course).

sCAP was defined as CAP with either at least one major criterion (NIV or IMV or HFNOT with ≥ 50% + PaO_2_/FiO_2_ ≤ 300 or septic shock with a need for vasopressors or blood pH < 7.30) or at least three minor criteria (respiratory rate ≥ 30 breaths/min, PaO_2_/FiO_2_ ≤ 250 mmHg, multilobar infiltrates, confusion, elevated blood urea nitrogen, WBC count ≤ 4 × 10^9^/L, platelet count ≤ 100 × 10^9^/L or > 400 × 10^9^/L, body temperature < 36 °C, hypotension requiring aggressive fluid resuscitation) [48,49]. ARDS was defined as acute onset or worsening of hypoxemic respiratory failure within 1 week of the estimated onset of the acute predisposing risk factor or new or worsening respiratory symptoms, not fully explained by cardiogenic edema or fluid overload with bilateral opacities on chest radiography and CT or bilateral B lines and/or consolidations on ultrasound not fully explained by effusions, atelectasis, or nodules/masses, with oxygenation criteria as defined by the new global definition of ARDS [50]. AKI was defined as an increase in serum creatinine of ≥26.5 µmol/L within 48 h or ≥ 50% within 7 days or urine output of <0.5 mL/kg/hour for >6 h [51].

As this was a non-interventional study, patients were treated according to the standard of care and at the discretion of the supervising physician.

All patients were monitored daily until discharge, and their clinical course and outcomes were noted (the need, level, and duration of oxygen therapy, the need and duration of renal replacement therapy, in-hospital mortality, 28-day mortality, hospitalization duration, ICU admission and days of ICU stay, development of complications).

### 4.3. Flow Cytometry Analysis of Lymphocyte Subpopulations

The data on CD3+ (T-lymphocytes), CD19+ (B-lymphocytes), absolute CD4+ T cell count, CD16/CD56 (NK cells), CD4+ (helper T cells), CD8+ (cytotoxic T-lymphocytes), CD8+CD38+ (activated CD8+ lymphocytes), and HLA-DR+ (activated T-lymphocytes) were collected from 22 patients with AdV pneumonia. These were measured using the BD FACSCanto II flow cytometer (Beckton Dickinson, Franklin Lakes, NJ, USA), and the results were analyzed using BD FACSCanto clinical software, version 4.0. For detection, the BD Multitest™ six-color TBNK reagent paired with BD Trucount™ tubes was used (Beckton Dickinson, Franklin Lakes, NJ, USA). This allows for the identification and quantification of T, B, and natural killer (NK) cells, as well as the CD4+ and CD8+ T cell subpopulations in peripheral blood, providing both percentage and absolute counts.

### 4.4. Analysis of Cytokine Expression

The serum levels of 13 cytokines (IL-4, IL-2, CXCL10 (IP-10), IL-1β, TNF-α, CCL2 (MCP-1), IL-17A, IL-6, IL-10, IFN-γ, IL-12p70, CXCL8 (IL-8), TGF-β1 (free active form) were measured in patients’ sera. Cytokine concentrations were determined using the Human Essential Immune Response Panel (13-plex) with a Filter Plate (Biolegend, San Diego, CA, USA), following the manufacturer’s protocol. Beads were incubated with 25 µL of serum samples, followed by the addition of a biotin-conjugated secondary antibody. The cytokines were detected using streptavidin-phycoerythrin, which binds to the biotin and emits a fluorescent signal. The analysis was performed using a BD FACSCanto II flow cytometer (Beckton Dickinson, Franklin Lakes, NJ, USA). The concentrations of the cytokines were calculated using LEGENDplex software (version 8.0, Biolegend, San Diego, CA, USA).

### 4.5. Statistical Analysis

Clinical characteristics, laboratory, and demographic data were evaluated and presented deceptively. Fisher’s exact test and Mann–Whitney U test were used to compare the two groups. For the comparison of cytokine concentrations among three groups, the Kruskal–Wallis test of variance with Dunn’s multiple comparisons test was used. Repeated measures two-way ANOVA test with Tukey’s multiple comparisons test was used to analyze the kinetics of laboratory markers between two groups at two times points (day 1 and day 5 of hospitalization). ROC analysis was performed to determine cut-off values of serum concentrations of laboratory markers at admission for differentiating patients with severe from non-severe CAP. Statistical analyses were performed using GraphPad Prism Software version 10 (San Diego, CA, USA).

## 5. Conclusions

Adenoviruses are important causes of CAP that predominantly affect immunocompetent young males. They are clinically and radiologically indistinguishable from bacterial CAP, associated with elevated inflammatory markers (CRP and procalcitonin), lymphopenia, mild thrombocytopenia, and liver injury, which are associated with pneumonia severity. Despite the prolonged clinical course and lack of response to antibiotics, they have generally good outcomes. Our findings might have several important practical implications: (1) they highlight the importance of viral testing in all hospitalized patients with CAP to identify patients with AdV; (2) prolonged fever and persistence of increased inflammatory markers despite antimicrobial therapy is common in AdV CAP and is rarely a sign of bacterial superinfection; therefore, antibiotic escalation in this setting is not warranted; (3) AdV detection in patients with CAP could encourage clinicians to deescalate or stop antimicrobial therapy; therefore, routine viral testing could be an important antimicrobial stewardship tool; (4) AdV CAP is associated with increased expression of proinflammatory cytokines and altered lymphocyte responses, which might be AdV-specific and differ from influenza, bacterial pneumonia or SARS-CoV2. This warrants further research to identify new immunomodulatory approaches for this group.

## Figures and Tables

**Figure 1 antibiotics-14-00023-f001:**
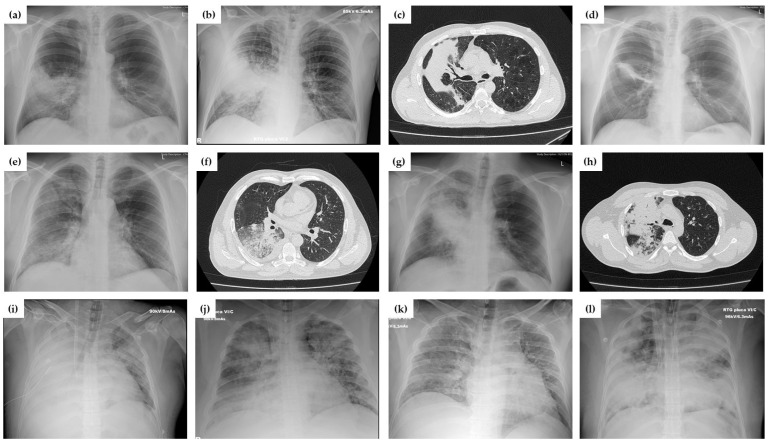
Selected radiograms and CT scans showing patterns and course of AdV CAP. Chest x-rays showing consolidation in the left lobe on day 1 (**a**) and further progression on day 4 (**b**) of hospitalization in a 40-year-old male treated with HFNOT. On day 15, CT scan (**c**) showed consolidation with interlobular septal thickening and air bronchogram, and at follow-up (on day 45), nodular infiltrate was still present (**d**). A 35-year-old male with non-severe CAP had a combination of consolidation with interstitial changes in the right lung (**e**) and on CT scan consolidation with ground–glass opacity and pleural effusion in the right lobe (**f**). A 31-year-old patient treated with HFNOT with unilateral consolidation on chest X-ray (**g**) and CT scan (**h**). Chest X-rays of patients (female, age 29; male, age 36; male, age 39; female, age 39) with bilateral infiltrates, pleural effusions, and ARDS, treated with vvECMO (**i**–**l**).

**Figure 2 antibiotics-14-00023-f002:**
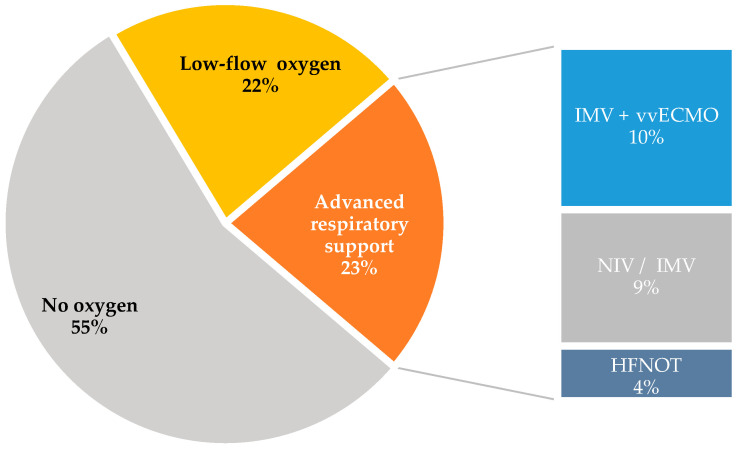
Maximal oxygen support during hospitalization.

**Figure 3 antibiotics-14-00023-f003:**
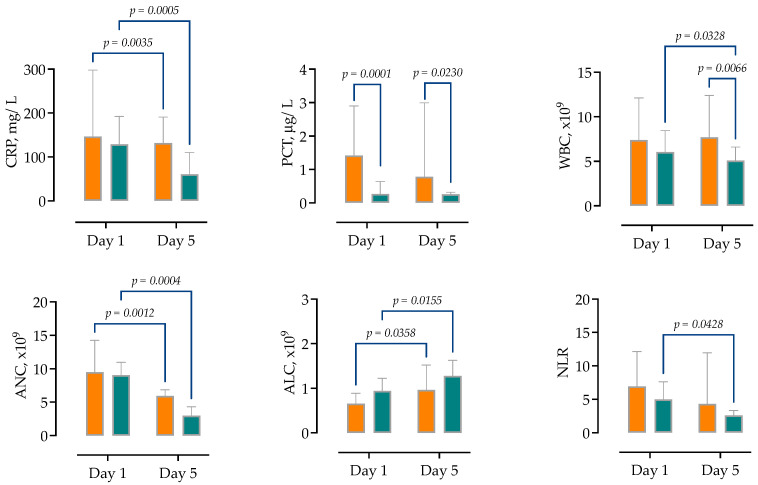
Comparison of routine laboratory markers at two selected time points (day 1 and day 5 of hospitalization), stratified by the severity of pneumonia. Shown are medians with IQRs, in orange color sCAP and in green non-severe CAP. Repeated measured two-way ANOVA with Tukey’s multiple comparisons test was used to calculate the source of variations. Abbreviations: CRP—C-reactive protein, PCT—procalcitonin, WBC—white blood count, ANC—absolute neutrophil count, ALC—absolute lymphocyte count, NLR—neutrophils-lymphocytes ratio.

**Figure 4 antibiotics-14-00023-f004:**
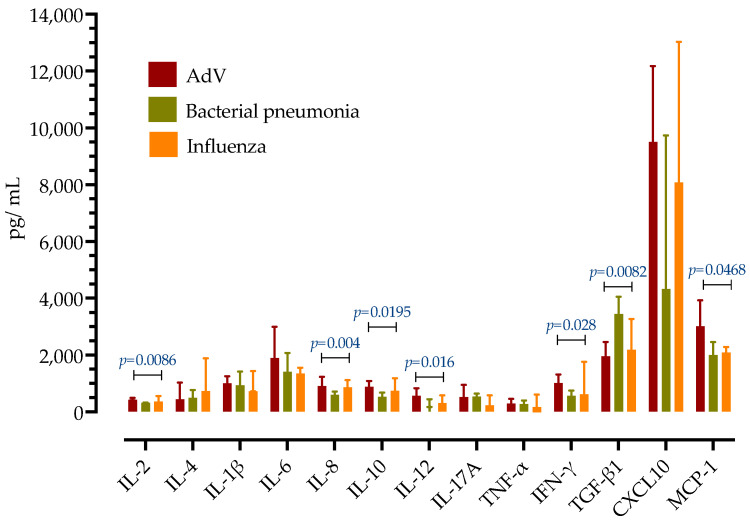
Serum concentrations of cytokines in patients with AdV pneumonia compared to influenza and bacterial pneumonia. Shown are medians with IQRs. *p*-values were calculated by ANOVA test.

**Figure 5 antibiotics-14-00023-f005:**
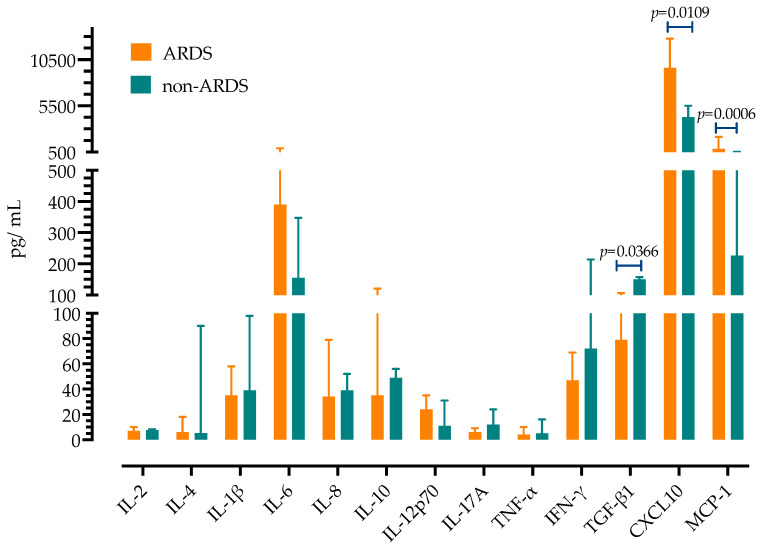
Association of cytokines with pneumonia severity and presence of ARDS. Shown are medians with IQRs with *p*-values calculated by Mann–Whitney U test.

**Table 1 antibiotics-14-00023-t001:** Comparison of routine laboratory findings at admission between patients with severe and non-severe AdV CAP.

	Non-Severe CAP (*n* = 36)	Severe CAP (*n* = 22)	*p*-Value *
Age, years, median (IQR)	32 (26–39)	38 (31–46)	0.0585
Male sex, *n* (%)	29 (80.6%)	18 (81.8%)	1.0000
Comorbidities, *n* (%)			
Diabetes mellitus	1 (2.8%)	1 (4.5%)	1.0000
Arterial hypertension	7 (19.4%)	3 (13.6%)	0.7269
Dyslipidemia	1 (2.8%)	2 (9.1%)	0.5508
Cardiovascular diseases	1 (2.8%)	1 (4.5%)	1.0000
Chronic kidney disease	1 (2.8%)	0	1.0000
Chronic obstructive pulmonary disease	3 (8.3%)	0	1.0000
Neurological diseases	2 (5.6%)	2 (9.1%)	0.6298
Smoking	14 (38.9%)	6 (27.3%)	0.4082
Moderate alcohol consumption	9 (25.0%)	4 (18.2%)	0.7475
Clinical presentation at admission			
Duration of illness at admission	6 (4–7)	8 (7–10)	<0.0001
Fever	36 (100%)	22 (100%)	1.0000
Peak temperature, °C	39.6 (39.3–40.0)	39.9 (39.1–40.1)	0.7923
Catarrhal symptoms	13 (36.1%)	7 (31.8%)	0.7833
Sore throat	18 (50.0%)	6 (27.3%)	0.1064
Diarrhea	25 (69.4%)	15 (68.2%)	1.0000
Dry cough	18 (50.0%)	17 (77.3%)	0.0542
Tachypnea	12 (33.3%)	16 (72.7%)	0.0063
Dyspnea	8 (22.2%)	16 (72.7%)	0.0003
Respiratory frequency, min	18 (16–22)	25 (24–29)	<0.0001
SpO_2_/FiO_2_	452 (438–461)	149 (93–225)	<0.0001
Antibiotic therapy before admission	21 (58.3%)	13 (59.1%)	1.0000
Duration of antibiotic therapy, days	3 (2–5)	3 (1–6)	0.7759
Clinical severity at admission			
SIRS score on admission	2 (2–3)	3 (2–4)	0.0065
CURB-65 on admission	0 (0–0)	2 (0–3)	<0.0001
PSI	42 (30–51)	74 (56–116)	<0.0001
SMART-COP	0 (0–1)	4 (2–5)	<0.0001
SOFA score	1 (0–2)	3 (2–7)	<0.0001

* Data are presented as frequencies (%) or medians (interquartile ranges). Fisher’s exact or the Mann–Whitney U test was used, as appropriate.

**Table 2 antibiotics-14-00023-t002:** Comparison of routine laboratory findings at admission between patients with severe and non-severe AdV pneumonia.

	Non-Severe CAP (*n* = 36)	Severe CAP (*n* = 22)	*p*-Value *
Laboratory findings on admission			
CRP, mg/L	129 (86–192)	147 (126–298)	0.1186
Procalcitonin, µg/L	0.27 (0.13–0.64)	1.4 (0.66–2.9)	0.0001
White blood cells, ×10^9^/L	6.1 (5.0–8.5)	7.4 (4.3–12.0)	0.2647
Neutrophils, %	74 (64–82)	78 (76–88)	0.0266
Lymphocytes, %	15 (10–20)	12 (7.1–15)	0.0766
Monocytes, %	8.5 (6.2–12)	6.4 (3.0–11.0)	0.0876
Neutrophils-lymphocytes ratio	5.0 (3.1–7.6)	6.9 (5.3–12)	0.0354
Lymphocyte-monocyte ratio	1.6 (1.3–2.2)	1.4 (1.1–3.3)	0.4878
Platelets, ×10^9^/L	143 (116–162)	137 (109–191)	0.7522
Platelet-lymphocyte ratio	167 (127–217)	207 (152–312)	0.1132
Hemoglobin, g/L	139 (132–144)	132 (120–152)	0.5822
Fibrinogen, g/L	5.9 (5.2–6.6)	5.5 (5.1–5.9)	0.1978
INR	0.96 (0.92–0.98)	0.94 (0.90–0.96)	0.2529
D-dimer, mg/L	1.6 (0.65–2.1)	3.5 (2.2–4.4)	0.0002
Glucose, mmol/L	5.9 (5.5–6.6)	7.6 (5.7–11)	0.0548
Blood urea nitrogen, mmol/L	4.2 (3.0–4.9)	7.9 (4.9–16.0)	<0.0001
Creatinine, μmol/L	79 (61–92)	111 (77–169)	0.0017
Na, mmol/L	136 (132–138)	137 (131–144)	0.3303
Total bilirubin, μmol/L	8.5 (6.0–10)	9.0 (7.0–11)	0.5586
AST, IU/L	37 (29–66)	143 (67–186)	<0.0001
ALT, IU/L	31 (22–41)	48 (32–99)	0.0028
GGT, IU/L	40 (22–58)	72 (38–119)	0.0062
ALP, IU/L	54 (44–74)	55 (49–78)	0.4995
LDH, IU/L	276 (199–372)	794 (473–1127)	<0.0001
CK, IU/L	219 (113–581)	923 (711–3246)	<0.0001
Serum albumins, g/L	38 (35–40)	33 (27–37)	0.0663

* Data are presented as medians with IQRs. Mann–Whitney U test was used to calculate *p*-values. Abbreviations: C-reactive protein (CRP); aspartate aminotransferase (AST); alanine aminotransferase (ALT); gamma-glutamyl transferase (GGT); international normalized ratio (INR); lactate dehydrogenase (LDH), creatine kinase (CK).

**Table 3 antibiotics-14-00023-t003:** Lymphocyte subpopulations in the peripheral blood of patients with AdV pneumonia.

	Non-Severe CAP	Severe CAP	*p*-Value *
Lymphocyte subpopulations			
CD3+ (T-lymphocytes), %	79 (45–80)	64 (59–71)	0.2678
CD19+ (B-lymphocytes), %	15 (10–47)	24 (16–30)	0.2398
Absolute CD4+ count/μL	829 (410–1192)	254 (167–427)	0.0194
CD16/CD56 (NK-cells), %	5.4 (2.9–13.0)	8.2 (5.5–12.0)	0.5610
CD4+ (helper T cells), %	40 (23–56)	38 (27–47)	0.7798
CD8+ (cytotoxic T-lymphocytes), %	28 (11–32)	23 (18–27)	0.8544
CD38+ (activated CD8+ lymphocytes), %	6.7 (5.0–15)	14 (9.8–18)	0.1186
HLA-DR+ (activated T-lymphocytes), %	13.0 (4.7–15)	6.8 (4.5–11)	0.3980
CD4+/CD8+ ratio	1.6 (1.2–3.2)	1.6 (1.3–2.1)	0.5884

* Data are presented as medians with IQRs. Mann–Whitney U test was used to calculate *p*-values.

## Data Availability

The datasets generated during and/or analyzed during the current study are available from the corresponding author upon reasonable request.

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
