# Peer review of "Adenoviral Pneumonia Outbreak in Immunocompetent Adults—A Missed Antimicrobial Stewardship Opportunity?"

_antibiotics, 2025, doi:10.3390/antibiotics14010023_

Round 1
Reviewer 1 Report
Comments and Suggestions for Authors
The manuscript title "Adenoviral pneumonia outbreak in immunocompetent adults - is withholding antibiotics an option?" is well-structured and provides valuable insights into AdV pneumonia management, particularly regarding antibiotic stewardship.
I have a few suggestions for the author:
1. Please correct the label in Figure 3.
2. This finding presents factors or laboratory tests that show some interesting data in AdV-infected patients. Given your findings, can you provide a strong interpretation of each result or conclusion to guide physicians in making decisions about antibiotic use and reduction of overuse in patients? For example, which chemistry, hematology, or inflammatory markers should be considered in conjunction with clinical presentation to decision antibiotic use?
3. Considering the limitation of single serotype detection in this study, could you discuss the potential implications of infection with different AdV serotypes? This would help readers better understand this aspect of the research.
Author Response
The manuscript title "Adenoviral pneumonia outbreak in immunocompetent adults - is withholding antibiotics an option?" is well-structured and provides valuable insights into AdV pneumonia management, particularly regarding antibiotic stewardship.
I have a few suggestions for the author:
Please correct the label in Figure 3.
Authors response: Corrected.
This finding presents factors or laboratory tests that show some interesting data in AdV-infected patients. Given your findings, can you provide a strong interpretation of each result or conclusion to guide physicians in making decisions about antibiotic use and reduction of overuse in patients? For example, which chemistry, hematology, or inflammatory markers should be considered in conjunction with clinical presentation to decision antibiotic use?
Authors response: We thank the reviewer for this comment. Although our study was not designed to examine the impact of antibiotic therapy on the outcomes of AdV pneumonia, the absence of detected bacterial coinfections, as well as the lack of clinical response to antibiotic therapy supports the early de-escalation or discontinuation of antimicrobial therapy. We have added this in the discussion section.
Considering the limitation of single serotype detection in this study, could you discuss the potential implications of infection with different AdV serotypes? This would help readers better understand this aspect of the research.
Authors response: We thank the reviewer for this observation. During this outbreak we only detected serotype AdV7. Published data on immunocompetent military recruits with serotypes 4, 7, 14 and 55, as well as a recent large study including 102 non-military immunocompetent AdV CAP adults (50% AdV-55 and 29% AdV-7), show comparable data. Still, this is an important limitation as stated in the Limitations section.
Reviewer 2 Report
Comments and Suggestions for Authors
In this manuscript, the authors analyze clinical trial data and laboratory findings to determine whether antibiotic treatment is necessary for AdV infection in immunocompetent patients. The paper requires significant reorganization and restructuring to present the information more clearly.
In the abstract, a clinical trial manuscript should begin with a clear statement of objectives: which treatments are being tested, in which patient populations, and with what expected outcomes? Currently, the first two sentences read more like an introduction highlighting AdV’s importance rather than outlining the study’s aims. Additionally, much of the results in the abstract are actually methods and should be moved accordingly. The conclusion section similarly includes new hypotheses, which is unconventional. A conclusion section should summarize the key findings and their implications, rather than introduce new speculation.
In the main section, the results need more coherent organization. The paper should explain why 58 patients were included and outline the rationale behind their grouping or selection criteria. It should state the underlying hypothesis tested within these groups and clarify the treatment arms and the presence or absence of a natural control group. Figure 1, for instance, focuses on a few patients out of the total 58, but the manuscript does not justify why these particular patients are shown or what insight they provide. Lines 180 to 195 feel off-topic and may need to be integrated more naturally or omitted. A structured, conventional flowchart could help illustrate the hierarchy of patient demographics, treatment options, and study design.
Finally, the manuscript should better connect laboratory results to clinical outcomes. It needs clearer explanations of what tests are considered conventional, what new observations were made, and how each piece of data supports the study’s conclusions.
Author Response
In this manuscript, the authors analyze clinical trial data and laboratory findings to determine whether antibiotic treatment is necessary for AdV infection in immunocompetent patients. The paper requires significant reorganization and restructuring to present the information more clearly.
- In the abstract, a clinical trial manuscript should begin with a clear statement of objectives: which treatments are being tested, in which patient populations, and with what expected outcomes? Currently, the first two sentences read more like an introduction highlighting AdV’s importance rather than outlining the study’s aims. Additionally, much of the results in the abstract are actually methods and should be moved accordingly. The conclusion section similarly includes new hypotheses, which is unconventional. A conclusion section should summarize the key findings and their implications, rather than introduce new speculation.
Authors’ response: We thank the reviewer for this comment and suggestions. This was a non-interventional, observational study that included all hospitalized patients with CAP with AdV detected in respiratory samples. The primary aim was to describe clinical, laboratory, radiographic and immunological characteristics of this patient group. Although this was not interventional study designed to examine the impact of antibiotic therapy od AdV CAP outcomes, we believe that our results support the early de-escalation or stopping of antimicrobial therapy. In addition, we provide a comprehensive description of prolonged clinical course and laboratory findings which mimic bacterial pneumonia and could encourage physicians to withhold antibiotic escalation.
We realize that the title might be misleading, therefore we changed it in “Adenoviral pneumonia outbreak in immunocompetent adults – a missed opportunity for antimicrobial stewardship”.
In addition, we changed the abstract accordingly to highlight its non-interventional design.
In the main section, the results need more coherent organization.
- The paper should explain why 58 patients were included and outline the rationale behind their grouping or selection criteria.
Authors’ response: As previously mentioned, the cohort consisted of all consecutively hospitalized patients with AdV pneumonia (total of 58), and this is now highlighted in the Methods and Results – Baseline patients’ characteristics section.
- It should state the underlying hypothesis tested within these groups and clarify the treatment arms and the presence or absence of a natural control group.
Authors’ response: We have now clarified study design in the Methods sections and further highlighted in the study limitations.
- Figure 1, for instance, focuses on a few patients out of the total 58, but the manuscript does not justify why these particular patients are shown or what insight they provide.
Authors’ response: We thank the reviewer for this suggestion. The idea was to show characteristic radiographic findings of AdV pneumonia which are radiographically indistinguishable from bacterial pneumonia, to help readers visualize AdV radiographic findings. We have added this explanation in the text.
- Lines 180 to 195 feel off-topic and may need to be integrated more naturally or omitted.
Authors’ response: We agree with the reviewer and in revised version we have omitted ROC-analysis. - A structured, conventional flowchart could help illustrate the hierarchy of patient demographics, treatment options, and study design.
Authors’ response: We thank the reviewer for this comment. However, as we previously described the non-interventional character of our study, we do not think the “flow chart” is needed, especially due to the length and comprehensiveness of the manuscript.
- Finally, the manuscript should better connect laboratory results to clinical outcomes. It needs clearer explanations of what tests are considered conventional, what new observations were made, and how each piece of data supports the study’s conclusions.
Authors’ response: We thank the reviewer for this comment. We have added this in the Conclusion section.